# Exploring the Impact of a Telehealth Care System on Organizational Capabilities and Organizational Performance from a Resource-Based Perspective

**DOI:** 10.3390/ijerph16203988

**Published:** 2019-10-18

**Authors:** Chun-Hsun Chen, Yu-Li Lan, Wei-Pang Yang, Fang-Ming Hsu, Chin-Lon Lin, Hsing-Chu Chen

**Affiliations:** 1Department of Business Administration, National Dong Hwa University, Executive Officer Office, Buddhist Tzu Chi Medical Foundation, Hualien 97401, Taiwan; junxun26@gmail.com; 2Department of Health Administration, Tzu Chi University of Science and Technology, Hualien 970, Taiwan; 3Department of Information Management, National Dong Hwa University, Hualien 97401, Taiwan; wpyang@gms.ndhu.edu.tw (W.-P.Y.); fmhsu@gms.ndhu.edu.tw (F.-M.H.); 4Executive Officer Office, Buddhist Tzu Chi Medical Foundation, Hualien 97401, Taiwan; cllinmd@tzuchi.com.tw; 5Office of Superintendent, Hualien Tzu Chi Hospital, Hualien 97401, Taiwan; tz1049@tzuchi.com.tw

**Keywords:** resource-based, telehealth care system, organizational capabilities, organizational performance

## Abstract

This study explored the effects of information technology (IT) resources—in conjunction with IT infrastructure and organizational resources—on organizational capabilities and performance. The study further analyzed the mediating effect of organizational capabilities on the relationship between IT resources and organizational performance. A cross-sectional research design was adopted, and questionnaire copies were administered to senior care supervisors of Taiwanese day care centers, care institutions, and hospitals. In total, 328 valid questionnaire responses were obtained. The study results are summarized as follows: (1) A direct effect analysis revealed that IT infrastructure significantly affected service performance and financial performance; organizational resources significantly affected service performance but did not significantly affect financial performance. (2) A mediation model analysis indicated that organizational capabilities exerted a mediating effect on the relationship between IT resources and organizational performance. These results can serve as a reference for medical care organizations in developing strategies for reviewing internal IT resources, integrating internal and external capabilities, creating a competitive advantage, and boosting their performance.

## 1. Introduction

Due to population aging and the increasing demand for health care services because of chronic diseases, countries seek to improve the quality of medical care services by investing heavily in the development of advanced communication technologies, including health information technology (HIT), for such services. The use of HIT constitutes an essential strategy for improving care and cost efficiency [1,2,3,4,5]. Moreover, telematics can be used to extend institutional care services to a community, a process that is generally referred to as telehealth care (THC). THC supports care in a community by improving care accessibility and quality while reducing cost [6,7,8,9].

However, definitive evidence regarding the performance of THC systems has yet to be furnished [10]. The primary problem associated with the derivation of such evidence is that IT investment indirectly affects performance through crucial mediating variables [11]. The most likely mediating variable is organizational capabilities. Studies have demonstrated that IT investments must be integrated with other organizational resources to improve performance [12]. Research has also emphasized that IT resources can affect performance through organizational capabilities. To provide effective and continuous care services, organizations must effectively integrate processes and resources in different environments. Thus, to maximize the effectiveness of THC, it must be integrated with the internal and external IT resources of an organization according to a patient’s home care needs [13,14,15].

Several studies have demonstrated that the use of IT resources has a positive effect on performance [16,17,18,19]. Such studies have used data to understand the source of the effect of IT on the competitive advantage of an organization or industry. However, only a few studies have explored the relationship between THC IT resources and THC organizational performance from the perspective of medical institutions [20,21]. Because of the continual evolution of IT, Kohli and Grover [22] suggested further research on the discontinuity of IT value to examine its intangible value. Therefore, scholars have increasingly explored the effectiveness of IT in improving and integrating processes, particularly with regard to the conditions for and benefits of using IT in integrating processes from different industries [23,24].

Studies have used the resource-based view (RBV) to explore the effect of various organizational resources on performance [11,12,25,26,27]. In addition to emphasizing how organizations can derive greater value from IT investments, studies have suggested that they use advanced IT resources to establish their core competencies and for their corporate strategic functions [28,29]. According to the RBV, enterprise success is primarily derived from internal resources; such resources should be characterized by value, heterogeneity, and nonmobility [30]. Bharadwaj [25] specified three dimensions of IT resources: IT infrastructure, human IT resources, and IT-enabled intangibles. Subsequent research has suggested a fourth dimension: IT reconfiguration capabilities [31,32].

Studies [18,19,23,33,34] have investigated the effects of complementary resources on the basis of the organizational capacity dimension of the IT competency model proposed by [25]. According to such studies, results obtained from measuring the complementarity of integrated IT resources are consistent with the RBV; competitors face difficulties in replicating such resources because they constitute a unique set of organizational resources that are established over time and are related to an organization’s history, culture, and experience [23,30]. In addition, other studies have applied mediation models to probe the mediating role of organizational capabilities in the relationship between IT resources and organizational performance [12,16,23,35]. Organizational capabilities are attributes of organizations that address environmental changes and management challenges. Organizations may need the ability to handle customers or link providers. IT resources can improve organizational capabilities and thereby improve business performance. Specifically, the synergy between organizational capabilities and IT resources enhances organizational performance.

Because these two models have been widely used in performance-related research, this study applied a mediation model to explore the indirect effect of IT resources in THC applications on organizational performance through organizational capabilities. The RBV model can be used to link and investigate the relationship between resources and performance. This model has three main structures: business performance, organizational resources, and capabilities [18,36,37,38]. Financial and operational performance can generally be improved by appropriately leveraging organizational resources and capabilities. In an organizational context, capabilities reflect a portfolio of resources that enable improved financial and operational performance in an organization [39].

This study explored the effect of THC IT resources—along with THC IT infrastructure and organizational resources—on THC organizational capabilities and THC organizational performance. The study further analyzed the mediating effect of THC organizational capabilities on the relationship between THC IT resources and THC organizational performance. On the basis of the research objectives and literature review, this study developed its conceptual framework (Figure 1) and proposes several hypotheses:

**H1:** 
*THC information resources positively affect THC organizational performance.*


IT resources include IT infrastructure, IT management, and IT human resources [12,25]. Liang et al. [12] proved that IT resources can enhance organizational capabilities and thus improve business performance. Related studies have also shown that THC can improve the quality of care and reduce costs [1,40,41].

**H2:** 
*THC IT resources positively affect THC organizational capabilities.*


During the process of importing THC, IT staff must possess the ability to solve various problems in the importation process. Higher IT staff skill levels are associated with smoother procedures of information system import, maintenance, and update [41,42,43].

**H2a:** *IT infrastructure positively affects an organization’s internal capabilities*.

IT infrastructure includes information systems, network facilities, and computer equipment [44,45].

Studies have revealed that IT infrastructure can strengthen resources for internal control capabilities, strengthen cooperation between departments, and improve system and development capabilities. These are key drivers of business performance [12,18,33,46,47,48,49].

**H2b:** *IT infrastructure positively influences an organization’s external capabilities*.

In addition to internal integration capabilities, the quality, accessibility, and cost of medical care are crucial and are of maximum interest to patients. In the face of rapid environmental changes, care organizations typically set their strategic direction toward IT development, including the development of technology related to market, customer response, external partner cooperation, and information-sharing capabilities [50,51]. Buntin et al. [52] showed that HIT engenders a significant improvement in the safety of medical care. Studies have also demonstrated that HIT reduces complications and mortality [53,54]. Jones et al. [55] studied published research reports and concluded that clinical decision support systems have a positive effect on quality, safety, and efficiency. Several studies have demonstrated that HIT significantly improves patient safety, reduces drug misuse, improves surgical safety, and prevents diagnostic errors [56,57,58,59]. By improving record-keeping, communication, and collaborative care systems, HIT can help ensure that information is accurate and used as a basis for care decisions [51,60].

**H2c:** *An organization’s IT resources positively affect an organization’s internal capabilities*.

Organizational IT resources—such as IT planning, investment decisions, coordination, and control—enable managing heterogeneous IT components within an organization and translating them into organizational skills to create business value [37,44]. Although IT infrastructure provides the foundation, an organization’s IT management capabilities have a considerable effect on business performance [61]. Mithas et al. [47] found that information management capabilities play a major role in customer management, process management, and performance management capabilities.

**H2d:** 
*An organization’s IT resources positively affect an organization’s external capabilities.*


Accumulating IT expertise and management capabilities often requires time, and IT staff must adapt to new technologies selected in an organization [62]. Research has predominantly applied IT human capital and organizational knowledge to determine an organization’s overall ability to assess IT opportunities. Enterprise IT human resources can be effectively integrated with other IT resources to attain a high level of performance. The importance of IT human resources for honing an organization’s competitive advantage in the digital economy is increasing, primarily in response to rapid environmental changes. A company with professional IT human resources can effectively plan and integrate IT and enterprise processes and can develop applications that are faster, more reliable, and more cost-effective than those of its competitors. In addition, such a company can effectively communicate and cooperate with other departments, predict future needs, and create new and more valuable products and services. Faced with rapid market changes and continual advances in IT, IT staff in care organizations must continue learning and integrating IT resources with internal organizational resources as well as connecting external partners to respond to market demands; these procedures can positively affect an organization’s external capabilities [18].

**H3:** 
*THC organizational capabilities positively affect THC organizational performance.*


Organizations’ competitive advantage and economic benefits engendered by the use of IT are typically related to IT management capabilities [63]. Ravichandran and Lertwongsatien [43] indicated that when a company’s IT resources are controlled by a higher level of management, a higher level of support can be provided to such resources; this can in turn affect the company’s business processes, engendering effective product and service innovation. Sacristán (2013) [64] proposed that patient-centered care in THC entails the use of IT to develop patient-centered applications. Accordingly [64], THC enables patients to be more involved in care activities.

**H3a:** 
*An organization’s internal capabilities positively affect its service performance.*


The integration of IT resources and enterprise resources—including the alignment of IT with business processes and the management of IT updates—can enable efficient use of IT resources; this can thus enable companies to attain a higher level of performance than their peers do. THC can integrate different specialist technologies to provide relatively safe and effective care for patients; consequently, THC can accumulate and embed care services over time, thus increasing the value and uniqueness of care. It can also hone a care organization’s competitive advantage and augments its performance [65,66]. Empirical research conducted in Taiwan has revealed that the use of technological innovation in hospitals can significantly affect the hospitals’ performance metrics such as daily outpatient visits, daily emergency visits, and occupancy rate [5]. Accordingly, caregivers can use IT resources to develop internal capabilities, which can in turn affect organizational service performance.

**H3b:** 
*An organization’s internal capabilities can positively affect financial performance.*


Barua et al. [67] examined the effect of IT on mediator variables, and they revealed that IT can indirectly influence organizational performance through various processes. The effect of IT on performance may be described by a two-stage model, with one stage involving the effect of the service process between the company and the customer, and the other stage being linked to performance through the service process. The focus of investing in enterprise IT is to reduce operating costs while improving the quality, speed, and time to market of goods and services. IT can thus be leveraged to create and enhance value for customers. Moreover, IT can be used to effectively control resources, strengthen coordination between organizations, and establish external capabilities that respond to customer needs. In certain scenarios, IT capabilities can increase other metrics such as customer satisfaction [68,69] and market share [67]. Ravichandran and Lertwongsatien [43] indicated that IT can be applied to develop core competencies for business operations and boost market performance. The internal capabilities of care organizations that use IT resources can expand the scope of services and increase revenue, thus improving financial performance.

**H3c:** 
*An organization’s external capabilities positively affect service performance.*


THC can be used for home patient monitoring, thus reducing traffic time, emergency and hospitalization rates, and patient mortality; the use of THC can also improve quality of life [65,70]. Many studies have explored the benefits of remote care, emphasizing aspects of service performance such as cost-effectiveness, patient health outcomes, professional counseling, and response time [71,72]. Lamont et al. [73] demonstrated that care institutions using THC can register a significantly higher occupancy rate, net operating income, and market share compared with competing institutions.

**H3d:** 
*An organization’s external capabilities positively affect financial performance.*


Kim et al. [37] demonstrated that IT can change a company’s financial performance through changes in business processes. IT resources have initial effects on organizational processes and subsequent effects on financial performance. Through the integration of HIT resources, THC can expand an organization’s scope of services to render patients more accessible, establish favorable medical relationships with patients, reduce costs, improve care quality and accessibility, and create customer value, thereby increasing the organization’s income and financial performance [65,66].

## 2. Materials and Methods

### 2.1. Sample and Data Collection

This study applied a cross-sectional research design and included high-level supervisors (chief, assistant dean, or director of information management) of day care centers, care institutions, and hospitals in Taiwan. From 20 December 2015 to 15 January 2016, a letter was sent to the organizations through an electronic document system to explain to the organizations that the study was commissioned by the Ministry of Health and Welfare in order to understand the benefits of promoting remote care systems to medical care institutions. From January to August 2016, the telephone contact organization confirmed the willingness to fill out the questionnaire and collect data. The research design and questionnaire in this study were reviewed and passed by the Human Subject Committee. No conflict of interest at all.

### 2.2. Research Tools

A structured questionnaire design was adopted, and the research tools were compiled according to the relevant domestic and foreign literature. The questionnaire was divided into three parts.

The first part pertained to THC IT resources for remote care and involved questions on the following items: (1) IT infrastructure, comprising information systems, networks, and computer equipment [44,45]; and (2) organizational IT resources, comprising human resources, internal partner management, external partner management, and data and application management.

The second part pertained to THC organizational capabilities and involved questions on the following items [34,37]: (1) internal capabilities, comprising customer orientation, knowledge assets, and collaboration [31,34,37,47]; and (2) external capabilities, comprising quality, cost, and proximity [17,47,55,70].

The third part pertained to THC organizational performance and involved questions on the following items: (1) service performance, comprising time-savings, productivity, and efficiency [45,74]; and (2) financial performance, comprising income, market share, and return on investment [11,43,75].

The validity and reliability of the questionnaire are described as follows.

1. Validity

Six senior experts in the fields of remote care, medical management, nursing and public health, community long-term care, and hospital information systems were invited to assess the validity of the questionnaire content on the basis of the six aforementioned items; they assessed the clarity and appropriateness of the text describing the items. The item descriptions were modified according to the experts’ suggestions. The content validity index (CVI) was applied to evaluate the validity of the questionnaire items. An overall CVI of 0.915 was derived, demonstrating the questionnaire to have excellent validity.

2. Reliability

To ensure that the questionnaire items were highly consistent with respect to the measurement of their associated facets, a reliability analysis was conducted. An overall Cronbach’s α value of 0.925 was derived for the questionnaire; the Cronbach’s α value for each facet was between 0.88 and 0.94 (Table 1). These values demonstrate that the facets of the questionnaire had a high degree of internal consistency.

3. Construct validity

Confirmatory factor analysis (CFA) was conducted to evaluate the construct validity of the questionnaire. To achieve a favorable model fit level, the goodness-of-fit index (GFI), adjusted GFI (AGFI), and normed fit index (NFI) must be greater than 0.9 and the root mean square residual (RMR) must be less than 0.05 (Hair et al., 2010). The CFA results revealed the following:(1)For the first part of the questionnaire, namely remote care IT resources, the second-order CFA revealed the following: χ^2^ = 23.375, df = 17, GFI = 0.982, NFI = 0.988, AGFI = 0.963, and RMR = 0.017. Therefore, the overall model suitability was determined to be in line with the aforementioned criteria.(2)For organizational capabilities, the second-order CFA showed the following: χ^2^ = 60.049, df = 38, GFI = 0.967, NFI = 0.979, AGFI = 0.943, and RMR = 0.018. Accordingly, the overall model suitability was noted to be in line with the aforementioned criteria.(3)Finally, for the third part of the questionnaire, namely organizational performance, the second-order CFA revealed the following: χ^2^ = 32.599, df = 8, GFI = 0.968, NFI = 0.977, AGFI = 0.917, and RMR = 0.016. Hence, the overall model suitability was determined to meet the aforementioned criteria.(4)Based on these results, the scale items of this study have convergence validity.

4. Discriminant validity

In general, the explanatory power represented by each factor is significantly different from that represented by other factors [76]. Statistically, the difference between tests is typically verified using the difference between an original (unconstrained) model and a constrained model. The original model describes the release correlation between various model facets, whereas the constrained model describes a certain structure. The coefficient of the correlation between facets is equal to 1. When the degree of freedom is 1, if the difference between the chi-squared values obtained for the original model and the constrained model is greater than 3.84, then any two facets in this model have different validity levels [77]. In this study, the results showed that for any two facets in the test model, the chi-squared values of the three sets of test items were are all greater than 3.84. Therefore, the facets of the display mode had significant discriminant validity.

## 3. Results

A total of 419 institutions were willing to complete this research questionnaire. The number of valid questionnaires was 328. The average annual salary determined for the institutions was 20 years. Among the various care institutions, 48 were day care centers (14.6%), 147 were community maintenance institutions (44.8%), and 133 were hospitals (40.5%). Moreover, regarding the regional distribution of the institutions in Taiwan, 80 (24.4%) were determined to be located in northern Taiwan, 67 (20.4%) were in central Taiwan, 139 (42.4%) were in southern Taiwan, and 42 (12.8%) were in eastern Taiwan. Concerning agency ownership, among the institutions, 154 were public institutions (47%), 135 were private institutions (41.2%), and 39 were corporate entities (11.9%). Furthermore, of the institutions, 47.2% were determined to have fewer than 20 employees, 29.6% (130 institutions) did not have full-time employees responsible for health information systems, and 22% (72) had more than five employees responsible for health information systems.

First, this study conducted a direct effect analysis. Indices of the degrees of fit between this study’s theoretical model and the linear structural model were as follows (Figure 2): χ^2^ = 49.200, df = 37, GFI = 0.975, AGFI = 0.956, RMR = 0.010, and RMSEA = 0.032. Therefore, the data and theoretical model reached an appropriate degree of fit, verifying the rationality of the theoretical model of this study. To examine the mediating role of THC organizational capabilities, this study followed the example of Liang et al. (2010) by using two models to detect mediator variables. The first model—the direct model between IT infrastructure and performance—revealed that THC IT resources significantly affected service and financial performance. However, organizational resources significantly affected service performance but not financial performance. The reason for this result may be related to Taiwan’s lack of clear regulations on the THC system charging model. Table 1 presents the direct effect analysis results.

Second, the study conducted a mediating effect analysis. The second model was divided into two phases (Figure 3): In the first phrase, the effects of THC IT resources on internal and external capabilities were observed. Organizational IT infrastructure and IT resources significantly affected internal and external capabilities. Furthermore, this study examined the effect of internal capabilities on external capabilities, and the results showed that internal capabilities significantly affected external capabilities.

In the second phase, the study explored the effects of internal and external capabilities on performance. Internal capabilities significantly affected service performance and financial performance; remarkably, external capabilities significantly affected both service performance and financial performance. The study results demonstrated that THC organizational capabilities had a mediating effect on THC IT resources and THC organizational performance (χ^2^ = 93.940, df = 86, GFI = 0.966, AGFI = 0.946, RMR = 0.015, and RMSEA = 0.017).

Third, this study verified the proposed hypotheses. A THC organizational capability mediation model was used for hypothesis testing. The results indicated that THC IT resources had a significant effect on THC organizational capabilities (Table 2). IT resources had a significant effect on internal capabilities and external capabilities (β = 0.23 ***; β = 0.63 ***); accordingly, H2a and H2b were supported. Additionally, organizational resources had a significant effect on internal capabilities and external capabilities (β = 0.16 ***; β = 0.32 ***); hence, H2c and H2d were supported. Internal capabilities had a significant effect on service performance and financial performance (β = 0.45 ***; β = 0.27 **); therefore, H3a and H3b were supported. Furthermore, external capabilities had a significant effect on service performance and financial performance (β = 0.32 ***; β = 0.30 ***); accordingly, H3c and H3d were supported. This study also conducted an R^2^ test to investigate the explanatory power of the variables. The results revealed that internal ability exhibited 50% explanatory power, whereas external ability exhibited 70% explanatory power. Moreover, service performance was determined to have 53% explanatory power, and financial performance was observed to have 59% explanatory power. Accordingly, the explanatory power levels of variables exceeded 50%.

## 4. Discussion

In recent years, the Taiwanese government has promoted a pilot program for remote care systems. Nevertheless, the effects of such systems have yet to be determined. Therefore, research is required to determine the performance effects of remote care systems. The current study was conducted to fill this research gap. The study findings demonstrate that THC IT resources are associated with THC organizational performance and that THC organizational capacity exerts a mediating effect on the relationship between THC IT resources and THC organizational performance, thus supporting the findings in the literature [45,78,79]. This study confirms that THC IT resources have a positive effect on THC organizational capabilities; therefore, such resources constitute a crucial prerequisite for the successful implementation of remote care [25].

IT infrastructure must be integrated with other resources to develop THC organizational capacity; such infrastructure cannot function alone. The study results also confirm that THC IT resources can positively influence external capabilities through internal capabilities. This indicates that care organizations can establish internal capabilities by integrating internal resources with THC IT resources; consequently, they can establish external capabilities. This result is consistent with those of previous research on the effect of IT resources on organizational performance. On the basis of resource classifications of [80], Kim et al. [37] classified capabilities and resources into organizational, human, and hardware facilities; they defined IT capabilities as “organizations combined with their organizations, people, and entities. The ability of IT resources to create value for the organization.” When organizations integrate their IT capabilities with their resources to rapidly adapt to changing circumstances, they can establish competitive organizational capabilities. According to [25], IT-enabled intangibles can be divided into three parts: (1) customer orientation, (2) intellectual assets, and (3) collaboration. Liang et al. [12] confirmed that organizational competence is a crucial regulatory variable for the relationship between IT resources and firm performance. The value of organizational capabilities lies in the existence of increase in complementary resources; this is because competitors face difficulties in replicating the overall effect of organization capabilities. Therefore, a mediation model of organizational capabilities states that the effect of IT resources on improved organizational performance is mediated by improvements in an organization’s internal and external capabilities [12]. Internal capabilities are deployed from within the organization to respond to market needs and opportunities, helping organizations to research and develop reliable products and services while minimizing unnecessary costs. External capabilities focus on mastering market needs, establishing lasting customer relationships, understanding the competitive strength of competitors in the market, and having the ability to plan and change IT strategies and adopt appropriate measures. According to the preceding specifications, the internal and external capabilities of an organization must be integrated to attain superior performance.

The results of this study also show that internal capabilities and external capabilities exert a significant positive effect on service performance and financial performance, signifying that care organizations can use THC IT resources to form THC organizational capabilities in order to improve service performance and financial performance. This result is consistent with those of previous research indicating that THC IT resources can indirectly affect performance by improving quality and increasing productivity. In conclusion, remote care institutions can integrate organizational THC IT resources to form THC organizational capabilities and improve care quality, thereby gaining the trust of patients and improving service performance. The study also confirms that remote care institutions can enhance their financial performance by improving service performance. The study results are consistent with previously proposed concepts concerning the relationship between THC organizational capabilities and THC organization performance. The results are also consistent with those of studies on the effectiveness of remote care.

## 5. Conclusions

Numerous THC application services have emphasized external value, such as concern for quality, cost reduction, and medical care accessibility improvement; nevertheless, such services have failed to consider strategies for overall IT resource development and integration with organizational capabilities. The current study suggests that care providers first review an organization’s THC IT resources and integrate such resources with the organization’s internal and external capabilities. These capabilities can accumulate and embed care service practices over time, thus increasing the value of care. Unique products and services provide care organizations with a competitive advantage and boost their performance. Caregivers can provide patient-centered managed care through a remote care system, develop a knowledge management system, facilitate information sharing within an organization, establish a decision support system, and strengthen the organization’s resource integration capabilities to provide patients with high-quality, accessible, and low-cost care services.

With the popularity of IT applications, research has recommended strengthening the ability to integrate IT, management knowledge, customer orientation, and synergy–related resources. Through the integration of business areas, organizations can achieve the following strategic activities: (1) more effective IT integration with business processes; (2) conception and development of software that is more reliable, cost-effective, and efficient than the competition to support business operations; (3) relatively efficient communication and operation across all business units; and (4) timely prediction of future customer needs and creation of new added value ahead of competitors [34,37]. In addition, cooperative institution management can have a major effect on organizational capabilities. Collaboration with partner care institutions can promote caregivers and partners to implement a patient-centric shared mission and vision [80].

This study developed its conceptual framework and hypotheses on the basis of the literature and developed a questionnaire to measure the variables. The researchers endeavored to be objective and rigorous in the research process. However, because the scope of this research was on IT resources, IT infrastructure, organizational resources, organizational capabilities, and organizational performance, issues related to integration could not be summarized herein. Moreover, telecare is still in the pilot phase globally, and several challenges must be overcome before its implementation in business operations; for example, devising strategies for integrating care systems with payment systems and emerging Internet of things applications is challenging. Accordingly, further research can be conducted on the discussed topics and challenges to provide comprehensive knowledge and findings. In addition to the internal and external organizational capacity changes proposed in this study, future research can explore other mediating variables that may exert positive effects on organizational performance. Findings from such research can enable care institutions and government bodies to promote remote care systems and effectively make appropriate care decisions.

## Figures and Tables

**Figure 1 ijerph-16-03988-f001:**
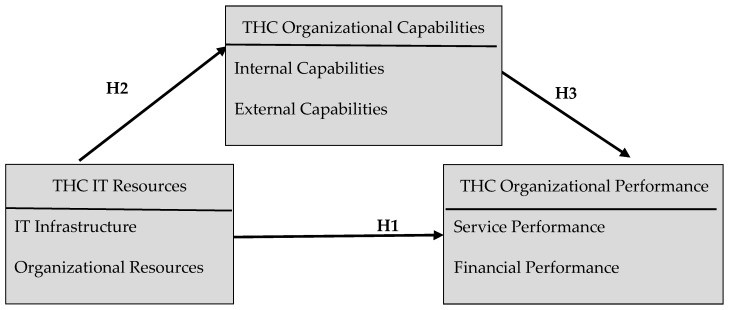
Conceptual framework. THC: telehealth care.

**Figure 2 ijerph-16-03988-f002:**
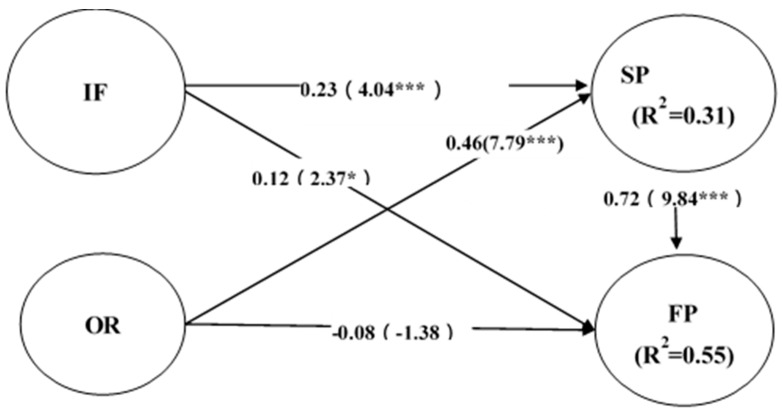
Direct effect analysis; *** *p* < 0.001; * *p* < 0.01.

**Figure 3 ijerph-16-03988-f003:**
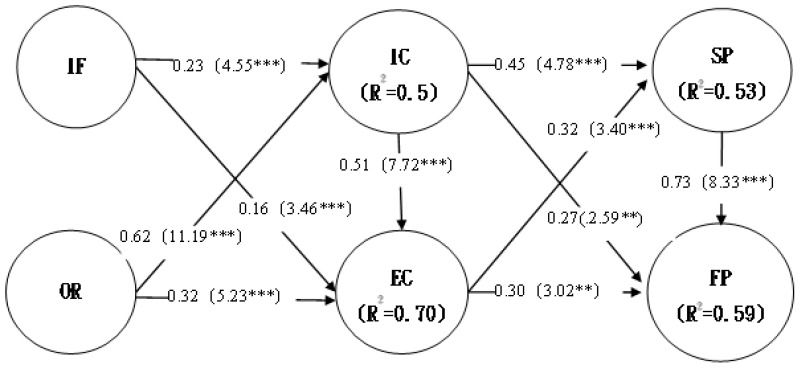
Mediation effect analysis; *** *p* < 0.0001; ** *p* < 0.001

**Table 1 ijerph-16-03988-t001:** Direct effect analysis results.

Mode			Mode (Direct Mode)		
Standard path coefficient/*p*-value			Standard path coefficient (normalized coefficient and *t* value)	Significance (*p*-value)	Outcome of Practice
Path/adaptation statistic					
Hypothesis	Hypothesis	Hypothesis			
H1a	IF → SP	**+**	0.23(4.04)	0.000	support
H1b	IF → FP	**+**	0.12(2.37)	0.019	support
H1c	OR → SP	**+**	0.46(7.79)	0.000	support
H1c	OR → FP	**+**	−0.08(−1.38)	0.166	Not support
Mode adaptation statistic					
Chi-square			49.200		
df			37		
GFI			0.975		
AGFI			0.956		
RMR			0.010		
REMSEA			0.032		

Abbreviations: IF: IT infrastructure; OR: organizational resources; SP: service performance; FP: financial performance; df: degree of freedom; GFI: goodness-of-fit index; AGFI: adjusted GFI; RMR: root mean square residual; REMSEA: root mean square error of approximation.

**Table 2 ijerph-16-03988-t002:** Mediation effect analysis results.

Mode			Mode 2 (Intermediary Mode)		
Standard path coefficient/*p*-value			Standard path coefficient (normalized coefficient and *t* value)	Significance (*p*-value)	Outcome of Practice
Path/adaptation statistic					
Hypothesis	Hypothesis path	Hypothetical relationship			
H2a	IF → IC	**+**	0.23(4.55)	0.000	support
H2b	IF → EC	**+**	0.62(11.19)	0.000	support
H2c	OR → IC	**+**	0.16(3.46)	0.000	support
H2d	OR → EC	**+**	0.32(5.23)	0.000	support
H3a	IC → SP	**+**	0.45(4.78)	0.000	support
H3b	IC → FP	**+**	0.27(2.59)	0.009	support
H3c	EC → SP	**+**	0.32(3.40)	0.000	support
H3d	EC → FP	**+**	0.30(3.02)	0.002	support
Mode adaptation statistic					
Chi-square			93.940		
d.f			86		
GFI			0.966		
AGFI			0.946		
RMR			0.015		
REMSEA			0.017		

Abbreviations: IF: IT infrastructure; OR: organizational resources; IC: internal capabilities; EC: external capabilities; SP: service performance; FP: financial Performance.

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
