# Peer review of "Exploring the Impact of a Telehealth Care System on Organizational Capabilities and Organizational Performance from a Resource-Based Perspective"

_ijerph, 2019, doi:10.3390/ijerph16203988_

Round 1

Reviewer 1 Report

In order to maximize THC effectiveness, I agree with the contentions that IT resources should be consolidated and organizational capacities should be formed. To this end, the author conducted the direct effect analysis and the mediation model analysis among IT resources, Organizational Capabilities and Organizational Performance. Clear introduction and conclusion are the strengths of this manuscript. However, more effort is required to reorganize the tables of results and clarify the wording. Additional descriptions that match the text and the results of the table are also required. In a minor point, it is hoped that the definitions of IT resources and IT infrastructure will be more clearly separated.

Author Response

審稿人評論

對評論的回應

需要付出更多的努力來重建結果表並消除措辭。

還需要與文本和表格結果相匹配的其他描述。

根據審閱者的意見,將結果表與文本匹配。

修訂後的內容包括:1 ..修訂後的內容與結果(圖和表)之間的一致性; 2.研究結果的補充說明。

(第306-307、309-310、312、327-329和331行)

在較小的一點上,希望將IT資源和IT基礎結構的定義更清楚地分開。

根據審閱者的意見,分別描述IT資源和IT基礎架構的定義。

(第110,119行)

Reviewer 2 Report

This study is to explore the effect of IT resources in organizational capabilities and organizational performance in health care settings. The study  analyzes the mediating effect of organizational capabilities on the relationship between IT resources and organizational performance health care settings.

Followings are some comments in order to improve the quality of the manuscript:

Add more health care and/or tele health care aspects in introduction. There are no such words between lines 46 and  107.  Add health care and/or tele health care or similar words in Conceptual framework. For example, THC IT Resources, THC Organizational Capabilities and THC Organizational Performance. Add THC in hypotheses. H1: Information resources positively affect organizational performance. => H1: Information resources positively affect organizational performance in telehealth care system. Apply it to others. Add data collection time period in 2.1. Sample and Data Collection. Add a sample size of number of supervisors participated in this study in 2.1. Sample and Data Collection.  In table 1, all languages should be in English. In table 1, 0.000 => 0.000. Since p value is provided, no *** necessary. Apply it to others. All Tables must be presented professionally.

Author Response

審稿人評論Reviewer’s Comments

對評論的回應

在簡介中添加更多醫療保健和/或遠程醫療保健方面。在第46和107行之間沒有這樣的詞。在概念框架中添加醫療保健和/或遠程醫療保健或類似的詞。例如,THC IT資源,THC組織能力和THC組織績效。tele health care aspects in introduction. There are no such words between lines 46 and  107.  Add health care and/or tele health care or similar words in Conceptual framework. For example, THC IT Resources, THC Organizational Capabilities and THC Organizational Performance.

根據審閱者的評論,在文本和概念框架中添加THC。(第49、83-86、96-107行)THC to text and conceptual framework. (Lines 49, 83-86, 96-107)

在假設中添加THC。假設1:信息資源對組織績效產生積極影響。=> H1:信息資源對遠程醫療系統中的組織績效產生積極影響。將其應用於其他人。THC in hypotheses. H1: Information resources positively affect organizational performance. => H1: Information resources positively affect organizational performance in telehealth care system. Apply it to others.

根據審閱者的評論,將THC添加到假設中。(第109、114、158行)THC to the hypothesis. (Lines 109, 114, 158)

在2.1中添加數據收集時間段。樣本和數據收集:在2.1中添加參與本研究的主管人數的樣本量。樣本和數據收集。

根據審閱者的意見,在2.1樣本和數據收集中添加描述樣本和數據收集時間段。(第209-216行)。此外,研究結果補充了參加研究的主管人數。(第287-288行)

在表1中,所有語言均應為英文。在表1中,0.000 => 0.000。由於提供了p值,因此無需***。將其應用於其他人。所有表格都必須專業呈現。

根據審閱者的意見,表單的內容已被修改。

修訂內容包括:

1.刪除表1和表2中的*,2.將表中的中文文本更改為英文,3.在圖2和圖3中為*添加註釋。

(第310、312、329、331行)
